# Using In Vitro Models to Study the Interactions Between Environmental Exposures and Human Microbiota

**DOI:** 10.3390/microorganisms13020247

**Published:** 2025-01-23

**Authors:** Qiwen Cheng, Shengxi Chen

**Affiliations:** 1Biodesign Center for Bioenergetics, Arizona State University, Tempe, AZ 85281, USA; 2Center for Big Data, Qilu Hospital of Shandong University, Jinan 250012, China

**Keywords:** human microbiota, in vitro models, environmental exposures, exposure-microbiota interactions

## Abstract

Research has demonstrated a close correlation between human microbiota and overall health, highlighting their intimate connection. Exposure to environmental factors, such as chemical contaminants and biological agents, has the potential to alter the composition and function of microbiota, thereby influencing health outcomes. Meanwhile, microbiota may contribute to host protection by degrading, or rendering harmless, exposures. Environmental exposures demonstrate significant diversity and dynamism; however, conventional methods for exposure–microbiota research, such as animal and epidemiological studies, are often both time-consuming and costly. Additionally, they may raise ethical concerns. This review aimed to examine the existing understanding of employing in vitro models to investigate the interactions between environmental exposures and human microbiota, particularly those located outside the large intestine. A comprehensive search was conducted across the Web of Science, PubMed, and Scopus databases, employing a range of keywords related to microbiota, exposures, and in vitro models. A total of 58 studies fulfilled the search criteria, revealing instances of microbial modulation of exposures and vice versa. It was observed that, although considerable research has been conducted on these interactions in vitro, there remains a pressing need for enhanced model designs and application contexts.

## 1. Introduction

Human microbiota, comprising trillions of microorganisms such as bacteria, archaea, fungi, and viruses, inhabit various body regions [1]. These microbial communities play critical roles in human health and disease. Gastrointestinal (GI) tract microbiota, particularly those in the large intestine, are crucial for nutrient metabolism, immune system development, and mental health [2,3]. Oral microbiota directly impact oral health and have been linked to conditions such as dental caries, periodontal diseases, and oral cancer [4]. Skin microbiota provide a defense against infections and inflammatory skin conditions, including acne, eczema, and psoriasis [5]. Vaginal microbiota create an acidic environment that inhibits the proliferation of opportunistic pathogens and prevents vaginal infections [6].

Imbalances within these microbial communities, often termed dysbiosis, can arise from environmental exposures. These exposures encompass chemical, biological, or physical agents that interact with humans and potentially lead to adverse health effects [7]. Exposure to chemical substances, such as heavy metals, pesticides, and antibiotics, can reduce microbial diversity, promote pathogen growth, and contribute to gut inflammation, increased intestinal permeability, and neurodevelopmental impairments [8,9,10,11,12,13]. Exposure to airborne pollutants, such as particulate matter, nitrogen dioxide, and ozone, can alter the composition and function of both respiratory and gut microbiota, increasing the risk of systemic inflammation and respiratory diseases [14,15]. Furthermore, contact with pathogens can interfere with microbial function, leading to either acute diseases or long-term health effects [1]. Despite these challenges, human microbiota possess remarkable defense mechanisms. Certain microorganisms can transform chemical toxins into less harmful forms, or bind and neutralize these toxins [16,17]. In addition, human microbiota can outcompete pathogens for nutrients and space, and create inhospitable environments by producing toxins or altering environmental pH [18,19]. Human microbiota can also modulate host immune responses, strengthen epithelial barrier integrity, and limit oxygen availability for facultative pathogens [18].

The continuous introduction of novel chemicals and the emergence of new pathogens underscore the importance of understanding the complex interactions between environmental exposures and human microbiota. Animal models, such as germ-free mice and traditional laboratory animals (e.g., rats), have played a crucial role in elucidating these interactions [20]. These models provide valuable insights into microbial and physiological responses to external stressors. Furthermore, genetically modified animal models enable researchers to explore specific host–microbiota interactions and the influence of host genetics on microbiota composition and function [21]. Nonetheless, the microbiota of laboratory animals may exhibit significant differences when compared to those of humans. For instance, animal vaginas lack key characteristics that are fundamental to the human vaginal environment, such as low pH and *Lactobacillus* dominance [22,23]. Such disparities can result in findings that may not be entirely applicable to human biology. Additionally, the ethical implications surrounding the treatment of animals in research, particularly in studies focusing on harmful exposures, raise important concerns that must be addressed [24,25,26]. Epidemiological studies offer another approach to directly investigating exposure–microbiota interactions in humans. Large-scale cohort studies can identify associations between exposures and health outcomes across diverse populations [27,28]. However, epidemiological studies often struggle to establish causal relationships, and it remains unclear whether changes in microbiota are a cause or consequence of disease. Additionally, human studies are subject to confounding factors like genetics, lifestyle, and socioeconomic status, which can complicate the interpretation of results.

To address these challenges, in vitro models have emerged as a valuable complementary approach. By enabling researchers to investigate exposure–microbiota interactions under controlled conditions, in vitro models circumvent the ethical and practical limitations of animal and human studies [24,25,26]. These models offer flexibility for high-throughput screening of individual or combined exposures, making them ideal for toxicity testing. Moreover, in vitro systems provide a stable and reproducible environment, facilitating the observation of sustained interactions without the complexities of animal or human studies. This review aimed to explore the role of in vitro models in elucidating the relationship between environmental exposures and human microbiota, with a particular focus on microbiota located outside the large intestine, an area that has received limited attention. An examination of the current in vitro models and the interactions between exposures and microbiota studied within these frameworks was conducted. Following this, the distinct advantages and drawbacks of existing research were analyzed, and recommendations for future efforts to enhance model application and tackle essential research inquiries were presented.

## 2. Methods

### 2.1. Search Strategy

The literature search was performed on 31 December 2024 across the Web of Science, PubMed, and Scopus databases, using a selection of keywords associated with oral, gastric, small intestinal, respiratory, skin, and vaginal microbiota, as well as exposures and in vitro models, in the abstract or title. A comprehensive list of query strings can be found in Appendix A.

### 2.2. Eligibility Criteria

The inclusion criteria comprised: (1) original research articles, theses, and dissertations; (2) in vitro studies examining exposure–microbiota interactions, which include changes in microbial growth, abundance, and activity due to exposure, as well as modulation of exposure chemical structure and availability by microbiota; and (3) publications written in English and Chinese.

The exclusion criteria were as follows: (1) studies not involving human microbiota (e.g., animal microbiota); (2) studies not related to environmental exposures (e.g., nutrients, drugs), with the exception of antibiotic drugs, due to the significant number of reports regarding unintended dietary exposure to antibiotic residues [29,30,31,32]; (3) studies not utilizing in vitro models; (4) non-original research (e.g., review articles, perspectives, awarded grants, errata, patents); (5) preprints; and (6) studies published in languages other than English and Chinese.

### 2.3. Data Extraction and Synthesis

The two authors independently assessed each study for eligibility. Subsequently, one author (Q.C.) extracted descriptive information from the full text of each eligible study, and synthesized exposure type(s), in vitro model type(s), key findings on exposure–microbiota interactions (including alterations in microbiota and/or exposures), and methods to identify key findings. The other author (S.C.) independently verified the accuracy of the results to reduce the risk of bias, and discussed any discrepancies with Q.C. 

## 3. Results

A total of 2147 records were retrieved from the three databases, with 1302 duplicate entries eliminated. The full texts of 16 out of the remaining 845 records were inaccessible and were therefore excluded from this review. Following the application of the inclusion and exclusion criteria, 58 studies were selected, which included 25 studies focusing on oral microbiota, one on gastric microbiota, one on small intestinal microbiota, seven on respiratory microbiota, 11 on skin microbiota, and 13 on vaginal microbiota (Appendix A). These studies were subsequently categorized into “gastrointestinal tract microbiota” and “extraintestinal microbiota” according to their respective locations, and are discussed in the following sections. In addition to these studies, other research was referenced mainly to establish a foundational context and to suggest potential avenues for future investigation, offering essential insights for comprehending this review.

### 3.1. Gastrointestinal Tract Microbiota

The microbiota in the human GI tract, including bacteria, archaea, fungi, and viruses, play a crucial role in human health [1,2,3]. These microorganisms inhabit various regions of the GI tract, from the oral cavity to the colon, and their composition and function can be influenced by factors such as environmental exposures. Research into GI tract microbiota has increasingly turned to in vitro models, which offer controlled environments for studying microbial dynamics and their interactions with exposures. The various large intestinal models and the types of exposures examined in vitro have been thoroughly reviewed in other literature; therefore, this work focuses primarily on non-large intestinal microbiota (i.e., microorganisms living in the oral cavity, stomach, and small intestine). For a comprehensive overview of large intestinal microbiota research, readers are referred to specialized reviews [8,9,10,11,12,13,20,24,25,26,33,34,35,36,37,38] (Table 1).

#### 3.1.1. Oral Microbiota

Oral microbiota, comprising over 1000 microbial species, are second in complexity only to the large intestinal microbiota [4,39]. Balanced oral microbiota contribute to dental health by preventing the overgrowth of pathogenic species that cause dental caries and periodontal diseases [4]. Multiple oral bacteria have also been linked to an increased risk of oral squamous cell carcinoma. Emerging research indicates a connection between oral microbiota and systemic health conditions, including cardiovascular diseases, diabetes, and respiratory infections [40]. A comprehensive understanding of oral microbiota can help inform strategies for maintaining oral and overall health, emphasizing the importance of good oral hygiene practices and regular dental care.

In vitro models designed to simulate oral microbiota range from simple single-species cultures to complex systems that closely resemble the physiological environment of the oral cavity. These models can be inoculated with either defined microbial species or actual samples obtained from human oral cavities, such as saliva, oral swabs, dental crowns, oral rinses, and toothbrushes [41,42,43,44,45,46,47,48,49,50,51]. Previous models aimed at investigating the interactions between oral microbiota and environmental factors encompass several innovative designs, including (1) the toothbrush model, which fosters biofilm development on nylon fibers (i.e., representative toothbrush material) [52]; (2) the hydroxyapatite disc biofilm reactor, which supports microbial biofilm growth on hydroxyapatite discs, effectively representing early supra-gingival plaques [42,53,54,55]; (3) the drip flow biofilm reactor, which allows a continuous drop-wise flow of medium over hydroxyapatite-coated slides, facilitating biofilms similar to supra-gingival plaques [42]; (4) the multiple sorbarod device, which enables the formation of biofilms similar to sub-gingival plaques [42]; and (5) the constant depth film fermenter, which simulates the oral environment by continuously supplying microbial media in a thin film of liquid flowing over the biofilm surface [43]. The last three models exhibit dynamic characteristics and are valuable for investigating the impact of mechanical forces on biofilms.

Previous research utilizing these models has primarily focused on examining the impact of oral hygiene products on microbial composition. For instance, fluoride, which is commonly utilized as an anti-caries agent in dental care products, showed minimal antimicrobial activity against biofilms consisting of *Candida albicans*, *Actinomyces oris*, *Fusobacterium nucleatum*, *Streptococcus oralis*, *Streptococcus sobrinus*, and *Veillonella dispar*, but decreased the formation of microbial extracellular polysaccharide (EPS) and the production of acids [54]. In a separate study, fluoride hindered the growth of *Streptococcus mutans* and *Streptococcus sanguinis* [44]. When used in conjunction with arginine, fluoride enhanced the growth of *S. sanguinis* and suppressed that of *S. mutans*, significantly reducing the demineralizing potential of oral biofilms derived from saliva. Other oral hygiene agents, including triclosan, chlorhexidine, traditional Chinese medicine, essential oils, and hypochlorite nanobubbles, could significantly reduce bacterial pathogens, including *Enterobacter*, *Streptococcus*, *Staphylococcus*, *Porphyromona*, and *Enterococcus*, as well as *C. albicans* in vitro [42,48,49,51,56,57,58,59,60]. However, it is important to note that certain agents might also hinder the growth of probiotics such as *Lactobacillus salivarius* and *Streptococcus salivarius* [51].

In addition to oral care products, previous in vitro research has linked oral microbiota to antibiotics and nanoparticles, two antimicrobial agents that can be unintentionally ingested. Studies on three antibiotics, namely, tetracycline, ampicillin, and amoxicillin, suggested that low concentrations of these antibiotics (e.g., less than 1 mg/L) had no impact or even a beneficial impact on the viability of oral biofilms, whereas elevated concentrations exhibited detrimental effects [43,45,46]. Furthermore, antibiotic resistance patterns within oral microbiota underwent significant changes following the introduction of antibiotics; however, the extent of these changes differed depending on the antibiotic types and concentrations, as well as the oral microbiota donors. Similar to antibiotics, nanoparticles possess the capability to modulate oral biofilm development. For instance, copper and zinc oxide nanoparticles could significantly reduce oral bacterial proliferation, EPS production, and biofilm formation [41].

Cigarette smoke and nonnutritive sweeteners are two additional chemical exposures linked to oral microbiota alterations in vitro. Cigarette smoking, associated with microbiota dysbiosis and periodontitis, could directly impact the abundance and function of *Fusobacterium*, a key player in oral biofilm development and disease progression [53]. Cigarette smoke could also enhance *S. mutans* biofilm formation [61]. Electronic cigarette vapor was capable of upregulating quorum sensing, enhancing oral biofilm surface area, increasing microbial alpha diversity, shifting metabolic pathways, and altering host response [55,62]. Nonnutritive sweeteners like acesulfame-K, aspartame, saccharin, and sucralose, were able to inhibit *S. mutans* and *S. sanguinis* biofilm development, reduce EPS production, lower *S. mutans*/*S. sanguinis* ratio, decrease acid production, and thereby lessen the cariogenic potential of oral biofilms [50].

Chemical factors are not the sole influences on oral microbiota; physical factors like radiation can also induce changes, as demonstrated in in vitro studies. For instance, low-dose gamma radiation (10 Gy) could reduce *Klebsiella oxytoca* biofilm formation, an effect potentially mitigated by the addition of mucins [63]. Additionally, heavy ion radiation could markedly reduce oral microbiota diversity, increase the relative abundance of *Streptococcus*, and upregulate *gtfC* and *gtfD* gene expression in *S. mutans*, indicative of enhanced cariogenic virulence [47].

The modulation of viral infections by oral microbiota is a burgeoning field of research. An in vitro assay for SARS-CoV-2 pseudovirus infection demonstrated that *Porphyromonas gingivalis* could significantly inhibit viral infection. This effect was mediated by *P. gingivalis*-related compounds, such as phosphoglycerol dihydroceramide and gingipains [64]. Similarly, in vitro studies involving *S. sanguinis* and Akata cells with Epstein-Barr virus (EBV) infection revealed that the metabolite of *S. sanguinis*, hydrogen peroxide, could induce EBV lytic activation [65].

#### 3.1.2. Gastric and Small Intestinal Microbiota

Despite its acidic environment, the stomach harbors a unique microbial community, though significantly less diverse than that in the oral cavity. Acid-resistant bacteria such as *Helicobacter pylori* and *Lactobacillus* species can colonize the stomach [66]. *H. pylori* infection can lead to gastritis, peptic ulcers, and even gastric cancer, while *Lactobacillus* can contribute to a healthy stomach by lowering pH, aiding digestion, and supporting the immune system [67,68]. Based on the literature search, continuous fermenters are the primary tools used to simulate the gastric environment [66,68], with pH as the only factor investigated in vitro [66]. It was found that when inoculated with gastric and duodenal aspirates and subjected to pH changes from 6.0 to 3.0, *Candida* and *Lactobacillus* species exhibited acid tolerance [66]. *Escherichia* and *Klebsiella* populations decreased with decreasing pH, though they persisted at significant levels at pH 3.0.

Relative to the acidic gastric environment, the small intestine’s more neutral pH facilitates diverse microbiota, composed of genera including *Lactobacillus*, *Bifidobacterium*, *Streptococcus*, *Enterococcus*, and *Escherichia* [69,70,71,72,73,74,75]. These bacteria aid digestion and nutrient absorption, regulate intestinal motility, and enhance mucosal immune function. Additionally, they may contribute to the gut–brain axis by producing metabolites like short-chain fatty acids (SCFAs), influencing mood and cognition [76]. Several recent investigations have sought to simulate small intestinal microbiota using batch cultures and in continuous reactors, the latter of which can function independently or be integrated into large intestinal models such as the Simulator of the Human Intestinal Microbial Ecosystem (SHIME) [69,70,71,72,73,74]. In one study, a dynamic in vitro model with four compartments, simulating the stomach, duodenum, jejunum, and ileum, was used to investigate and compare the impact of a bacteriophage cocktail and the antibiotic ampicillin on seven representative ileal microbial species and the foodborne pathogen *Listeria monocytogenes* [75]. While both treatments effectively inhibited *L. monocytogenes*, the bacteriophage cocktail demonstrated superior specificity, avoiding the microbiota dysbiosis-inducing effects associated with ampicillin.

**Table 1 microorganisms-13-00247-t001:** Interactions between intestinal microbiota and environmental exposures using in vitro methods.

Exposure	In Vitro Model	Key Findings ^1^	Methodology ^3^	Reference
Oral cavity microbiota				
Sodium fluoride	Six-species biofilm on sintered hydroxyapatite disks	Total bacteria (-)	Viable cell counting	[54]
Sodium fluoride	Saliva-derived mixed-species biofilm on saliva-coated human enamel discs	*Streptococcus mutans* (↓) *Streptococcus sanguinis* (↓)	qPCR	[44]
Stannous fluoride, triclosan + sodium fluoride	Saliva-derived mixed-species culture	Uncultured *Veillonella* sp. (↑) *Bulleidia extructa* (↑) *Veillonella atypica* and three *Veillonella* sp. (↓)	DGGE	[48]
Sodium fluoride + arginine	Saliva-derived mixed-species biofilm on saliva-coated human enamel discs	*Streptococcus mutans* (↓) *Streptococcus sanguinis* (↑)	qPCR	[44]
Sodium fluoride + stannous chloride	Oral isolate single-species culture	*Enterobacter hormaechei* (↓) *Streptococcus salivarius* (↓) *Staphylococcus aureus* (↓) *Enterobacter cloacae* (↓) *Enterococcus faecalis* (↓) *Lactobacillus salivarius* (↓) *Candida albicans* (↓)	Viable cell counting	[51]
Stannous fluoride + zinc lactate	Saliva-derived mixed-species biofilm in hydroxyapatite disc reactors	Total facultative anaerobes (↓) Total anaerobes (-) Total streptococci (-) Total Gram-negative anaerobes (↓)	Viable cell counting	[42]
Stannous fluoride + zinc lactate	Saliva-derived mixed-species biofilm in drip-flow biofilm reactors	Total facultative anaerobes (↓) Total anaerobes (↓) Total streptococci (↓) Total Gram-negative anaerobes (↓)	Viable cell counting	[42]
Stannous fluoride + zinc lactate	Saliva-derived mixed-species biofilm in multiple sorbarod devices	Total facultative anaerobes (-) Total anaerobes (-) Total streptococci (-) Total Gram-negative anaerobes (↓)	Viable cell counting	[42]
Triclosan	Saliva-derived mixed-species biofilm in hydroxyapatite disc reactors	Total facultative anaerobes (↓) Total anaerobes (↓) Total streptococci (↓) Total Gram-negative anaerobes (↓)	Viable cell counting	[42]
Triclosan	Saliva-derived mixed-species biofilm in drip-flow biofilm reactors	Total facultative anaerobes (↓) Total anaerobes (↓) Total streptococci (↓) Total Gram-negative anaerobes (↓)	Viable cell counting	[42]
Triclosan	Saliva-derived mixed-species biofilm in multiple sorbarod devices	Total facultative anaerobes (-) Total anaerobes (-) Total streptococci (↓) Total Gram-negative anaerobes (↓)	Viable cell counting	[42]
Traditional Chinese medicinal toothpaste	Oral cavity-derived isolate single-species culture	*Enterobacter hormaechei* (↓) *Streptococcus salivarius* (-) *Staphylococcus aureus* (↓) *Enterobacter cloacae* (-) *Enterococcus faecalis* (↓) *Lactobacillus salivarius* (-) *Candida albicans* (↓)	Viable cell counting	[51]
Chlorhexidine	Single-species culture and biofilm in culture plates; dual-species culture and biofilm in culture plates	*Streptococcus mutans* (↓) *Candida albicans* (↓) *Staphylococcus aureus* (↓) *Pseudomonas aeruginosa* (↓)	Viable cell counting	[57]
Chlorhexidine gluconate	Oral cavity-derived *Candida albicans* isolate single-species culture	*Candida albicans* (↓)	Cell counting, optical density measurement	[56]
Clove, oregano, thyme essential oils	Mixed-species biofilm in culture plates, and plates supplemented with nylon fibers	Mixtures of 5–6 species selected from *Actinomyces viscosus*, *Enterococcus faecalis*, *Streptococcus mutans*, *Streptococcus oralis*, *Streptococcus sanguinis*, and *Streptococcus salivarius* (↓)	Visual turbidity, viable cell counting, crystal violet staining	[52]
Two tulsi essential oils	Single-species culture on agar plates	*Porphyromonas gingivalis* (↓) *Prevotella intermedia* (↓) *Fusobacterium nucleatum* (↓) *Staphylococcus aureus* (↓) *Streptococcus mutans* (↓)	Agar well diffusion assay	[58]
Eleven essential oils from plant rhizome, leaf and bark	Single-species culture on agar plates	*Streptococcus mutans* (↓) *Streptococcus sanguinis* (↓) *Staphylococcus aureus* (↓) *Candida albicans* (↓)	Disc diffusion assay	[59]
Hypochlorite nanobubbles	Saliva-derived mixed-species culture	*Porphyromonas pasteri* (↓)	16S rRNA gene sequencing	[49]
Denture cleanser	Nine-species biofilm on polymethylmethacrylate discs	Total aerobes (↓) Total anaerobes (↓) *Candida* (↓)	qPCR	[60]
Copper oxide nanoparticles, zinc oxide nanoparticles	Teeth crown surface-derived mixed-species culture	Total bacterial counts (↓)	Viable cell counting	[41]
Tetracycline	Saliva-derived mixed-species biofilm in Constant Depth Film Fermenters	Total anaerobic count (↓) *Lactobacillus* (-) *Streptococcus* (↓) *Actinomyces* (↓)	Viable cell counting	[43]
Ampicillin	Saliva-derived mixed-species biofilm in culture plates pre-coated with saliva pellicle	*Veillonella atypica* (↑) *Veillonella infantium* (↑) *Veillonella dispar* (↑) *Veillonella parvula* (↓) *Prevotella jejuni* (↑) *Prevotella histicola* (↑) *Prevotella salivae* (↑) *Prevotella melaninogenica* (↑) *Streptococcus oralis* (↓) *Streptococcus mitis* (↓) *Streptococcus parasanguinis* (↓) *Streptococcus sanguinis* (↓) *Streptococcus salivarius* (↑) *Streptococcus pneumoniae* (-) *Staphylococcus aureus* (-)	Metagenomic shotgun sequencing	[45]
Amoxicillin	Saliva-derived mixed-species biofilm in culture plates	Total viable cells (-) *Streptococcus salivarius* (↑) *Streptococcus pneumoniae* (↑) *Lactobacillus fermentum* (↓)	Viable cell counting, metagenomic shotgun sequencing	[46]
Cigarette smoke	Mixed-species biofilm in sintered hydroxyapatite disc reactors	*Fusobacterium nucleatum* was associated with carbohydrate metabolism (↑), cofactors, vitamins, prosthetic groups and pigments (↑), amino acid metabolism (↑), virulence mechanisms (↑), respiration (↓)	Metatranscriptomic sequencing	[53]
Cigarette smoke	Single-species biofilm in culture plates	*Streptococcus mutans* biofilm (↑)	Crystal violet staining	[61]
Electronic cigarette vapor	Mixed-species biofilm on sintered hydroxyapatite disks, with or without organoid tissue overlay	Without overlay: quorum-sensing regulated gene expression (↑), biofilm surface area (↑) With overlay: keratin thickness (↑), host response to pathogen-rich biofilms (↓)	NMR spectroscopy, TIMS-TOF, CLSM	[55]
Electronic cigarette vapor	Saliva-derived mixed species culture in 3D oral mucosa models	Alpha diversity (↑) *Clostridium* (↑) *Prevotella* (↑) *Veillonellaceae* (↑) *Bacteroides* (↑) Multiple glucose and energy metabolic pathways (↑)	16S rRNA gene sequecning, GC-MS	[62]
Nonnutritive sweeteners including acesulfame-K, aspartame, saccharin, and sucralose	Single-species culture and biofilm in culture plates; dual-species biofilm on glass coverslips pre-coated with saliva; saliva-derived mixed-species biofilm on glass coverslips pre-coated with saliva	*Streptococcus sanguinis* (↓) *Streptococcus mutans* (↓) *Streptococcus mutans*/*Streptococcus sanguinis* ratio (↓)	Optical density meansurement, FISH, EPS staining	[50]
Gamma radiation	*Candida albicans*, *Candida glabrata*, *Streptococcus salivarius*, and *Klebsiella oxytoca* single-species culture and biofilm in culture plates	Planktonic cell growth (-) *Klebsiella oxytoca* and *Candida glabrata* biofilms exhibited varying responses to different culture conditions	Optical density meansurement, crystal violet staining, calcofluor white staining	[63]
Heavy ion radiation	Single-, dual-, and saliva-derived mixed-species culture	*Streptococcus* (↑) *Streptococcus mutans*/*Streptococcus sanguinis* ratio (↑)	Viable cell counting, qPCR, FISH, MTT assay, crystal violet staining, EPS staining, RNA sequencing	[47]
SARS-CoV-2	*Porphyromonas gingivalis*, *Actinobacillus actinomycetemcomitans, Actinomyces odontilyticus* single-species culture supernatant, co-cultured with ACE2 + 293 T cells	SARS-CoV-2 pseudoviral infection (↓)	Luciferase activity measurement	[64]
Epstein-Barr virus (EBV)	*Streptococcus sanguinis* and Akata cell co-culture	EBV lytic activation (↑)	Flow cytometry, qPCR	[65]
Gastric microbiota				
pH (6.0 to 3.0)	Eleven-species culture in chemostats	*Candida* (-) *Lactobacillus* (-) *Escherichia* (↓) *Klebsiella* (↓)	Viable cell counting	[66]
Small intestinal microbiota				
Bacteriophage cocktail	Seven-species culture in the Smallest Intestine (TSI) model inoculated with *Listeria monocytogenes*	*Streptococcus* (-) *Enterococcus faecalis* (-) *Listeria monocytogenes* (↓) *Escherichia coli* (-)	Viable cell counting	[75]
Ampicillin	Seven-species culture in the Smallest Intestine (TSI) model inoculated with *Listeria monocytogenes*	*Streptococcus* (-) *Enterococcus faecalis* (↓) *Listeria monocytogenes* (-) *Escherichia coli* (↓)	Viable cell counting	[75]
Large intestinal microbiota ^2^				
Reviews on types of in vitro models				[24,25,26,33]
Reviews including exposure-microbiota interactions using in vitro models:				
Heavy metals				[8,9]
Antibiotics				[10,11]
Nanomaterials				[34,35]
Persistent organic pollutants				[12,13]
Food additives				[36,37]
Pathogens				[20,38]

^1^ The symbols (↑), (↓) and (-) represent significant increases, decreases, and no significant changes in microbial growth, abundance, or activity, respectively, as observed after exposure. ^2^ Numerous reviews on large intestinal microbiota are already available, and a selection of these reviews is listed in this table. ^3^ qPCR: quantitative polymerase chain reaction; DGGE: denaturing gradient gel electrophoresis; FISH: fluorescence in situ hybridization; EPS: extracellular polysaccharide; MTT: 3-(4,5-dimethylthiazol-2-yl)-2,5-diphenyltetrazolium bromide; NMR: nuclear magnetic resonance; TIMS-TOF: trapped ion mobility spectrometry time-of-flight mass spectrometry; CLSM: confocal laser scanning microscopy; GC-MS: gas chromatography coupled to mass spectrometry.

### 3.2. Extraintestinal Microbiota

Beyond the GI tract, microbial communities reside in diverse body sites, including the respiratory tract, skin, and vagina, where they play critical roles in maintaining health and influencing disease outcomes. In vitro models have become a valuable tool for advancing the understanding of extraintestinal microbiota, a field that remains less explored compared to the well-studied intestinal microbiota. The literature search yielded only 31 studies investigating in vitro interactions between extraintestinal microbiota, including the respiratory tract, skin, and vaginal microbiota, and various environmental exposures (Table 2 and Appendix A). In contrast, 25 studies focused solely on oral microbiota, while hundreds examined the large intestinal microbiota. Therefore, a comprehensive overview of current research on extraintestinal microbiota and their interactions with exposures in vitro is presented, along with the identification of research domains that necessitate further exploration.

#### 3.2.1. Respiratory Microbiota

The respiratory tract harbors a diverse community of microorganisms, collectively known as respiratory microbiota. Key bacterial genera found in this ecosystem include *Staphylococcus*, *Corynebacterium, Streptococcus*, *Haemophilus*, and *Prevotella* [77,78,79]. These microorganisms play a crucial role in respiratory health by influencing immune responses and pulmonary function. Disruptions to this delicate balance can lead to various respiratory conditions, such as pneumonia, chronic obstructive pulmonary disease, and asthma [77,80]. Additionally, interactions between the gut and respiratory microbiota, often referred to as the gut–lung axis, can influence systemic immune responses and inflammation, impacting respiratory health [80].

Despite its importance, research into respiratory microbiota is still emerging. Current in vitro studies often rely on conventional culturing methods to investigate specific bacterial isolates, but a few more complicated models have been developed. For example, the simultaneous utilization of a filter plate alongside a standard multi-well plate (receiver) facilitates the examination of the effects of soluble microbial metabolites that can transfer from the filter plate to the receiver [81]. The nasal epithelial cell model allows microbial colonization of a cultured host mucosa in vitro, providing a platform for investigating the intricate dynamics of host–microbe and microbe–microbe interactions [82]. The three-dimensional (3D) lung epithelial model demonstrates the ability to replicate bacterial invasion and host pro-inflammatory response [82]. The air-liquid interface (ALI) culture model, often using cell lines like Calu-3, simulates the respiratory tract by creating an interface between air and liquid, allowing for the differentiation of cells into a functional, mucus-secreting epithelium, which can then be co-cultured with various respiratory microbiota [77].

Using microbial cultures, previous studies have isolated *Staphylococcus* [78,83], *Haemophilus* [84], *Prevotella* [79], *Moraxella catarrhalis* [85], and *Streptococcus pneumoniae* [85] from nasal or throat swabs and examined their antibiotic susceptibility on agar plates. Furthermore, one study cultured sputum microbiota in a batch model and investigated the impact of excessive oxygen on microbiota composition and function [86]. Results from this study indicated that hyperoxia reduced the overall microbial load and diversity, as well as the abundance of specific bacteria, including *Rothia mucilaginosa* and various *Streptococcus* species. In contrast, *Pseudomonas aeruginosa* and *Staphylococcus aureus*, which are commonly associated with cystic fibrosis, were minimally affected. In addition, the ALI model was used to study the impact of microbial changes in response to human rhinovirus (HRV) infection, and it was discovered that a combination of *Corynebacterium pseudodiphtheriticum* and *Haemophilus influenzae* significantly reduced HRV copy number, highlighting the potential protective role of these bacteria against viral infections [77].

#### 3.2.2. Skin Microbiota

Skin microbiota are a diverse community of microorganisms, including bacteria, fungi, viruses, and archaea, that reside on and within the skin. Common inhabitants include *Staphylococcus*, *Cutibacterium*, *Micrococcus*, *Propionibacterium*, *Corynebacterium*, and *Malassezia* species [5,87]. These microorganisms play a vital role in maintaining skin health by protecting against pathogens, regulating the immune response, and contributing to overall skin homeostasis. Balanced skin microbiota support wound healing, prevents infections, and reduces inflammation, while dysbiosis has been associated with dermatological conditions such as acne, eczema, psoriasis, and atopic dermatitis [5].

Currently, the majority of in vitro studies on skin microbiota rely on traditional culturing techniques, whereas advanced models designed to replicate the physical architecture and function of the skin do not typically integrate the microbial ecosystem. According to our knowledge, a limited number of studies have attempted to inoculate skin microorganisms into skin models, with a maximum of two species being introduced simultaneously [87,88]. One of these models, the microbially competent 3D skin model, was utilized to study the impact of polycyclic aromatic hydrocarbons, such as benzo[a]pyrene (B[a]P), on skin microorganisms *Micrococcus luteus* and *Pseudomonas oleovorans* [87]. This research demonstrated that B[a]P had the potential to function as the exclusive carbon and energy source for the two microorganisms, with its metabolites experiencing modified rates of skin penetration and diffusion.

In single-species cultures, the most-studied environmental factor affecting skin microbiota is the use of skincare and cosmetic products. The presence of diverse ferments and plant extracts in cosmetics could influence the production of SCFAs by *Staphylococcus epidermidis* [89]. Additionally, certain sunscreen components, including butyl methoxydibenzoylmethane, ethylhexyl salicylate, and octocrylene, along with their combinations, were able to enhance the viability of the probiotic *Lactobacillus crispatus* while diminishing the presence of the pathogenic *Cutibacterium acnes* upon UV exposure [90]. Conversely, skin microorganisms such as *Deinococcus grandis* and the genus *Stenotrophomonas* were capable of metabolizing some of these ingredients [91]. In addition to chemical products, ultraviolet radiation (UVR) is a common skin exposure. Under UVR, *Sphingomonas mucosissima* demonstrated significant resistance and the ability to lower reactive oxygen species levels in human keratinocyte cell lines, suggesting its potential role in safeguarding human skin from UV-induced damage [92]. In addition, UVA strongly inhibited the growth of *M*. *luteus*, *Corynebacterium stearicum*, *Moraxella osloensis*, and a few *Staphylococcus* species, while UVB only delayed the exponential phase or showed insignificant impact [93]. In a separate study, UVB substantially inhibited the proliferation of *C. acnes* and *Lactobacillus fermentum* [94].

Other forms of exposure have also been explored in vitro. Mycolactones produced by *Mycobacterium ulcerans*, the causative agent of Buruli ulcer, could significantly stimulate spore germination of *Aspergillus flavus* and *Aspergillus niger* while inhibiting *Penicillium rubens*, highlighting intricate interactions between mycobacteria and fungi [95]. Moreover, two types of azo dyes, namely methyl red and orange II, were effectively reduced by skin bacteria such as *Staphylococcus*, *Micrococcus*, and *Kocuria* [96]. Finally, antimicrobial agents, such as antibiotics and green tea extracts, were co-cultured with prevalent skin microorganisms to determine their inhibitory effects [97,98].

#### 3.2.3. Vaginal Microbiota

Vaginal microbiota are essential for sustaining both vaginal and reproductive health, serving as a defense against infections and impacting a range of health outcomes [99,100]. Healthy vaginal microbiota are predominantly composed of *Lactobacillus* species, which foster an acidic environment via production of lactic acid, thereby inhibiting the proliferation of pathogens. Key species within this group include *L. crispatus*, *L. jensenii*, and *L. gasseri*, though the dominance of these species can vary among individuals [100,101]. An imbalance in vaginal microbiota has been linked to conditions like bacterial vaginosis (BV) and an increased risk of sexually transmitted infections [99,102,103,104], highlighting the importance of this microbial community in maintaining health.

In vitro investigations concerning vaginal microbiota primarily depend on microbial cultures, with a few models established within more intricate systems. For instance, in an ALI culture model, vaginal bacteria and vaginal epithelial cells were co-cultured to mimic the morphological and functional characteristics of the vaginal mucosa and the production of microbial metabolites, as well as viral infection [100,102,105]. In a vagina-on-a-chip microfluidic model, the probiotic *L. crispatus* and disease-associated *Gardnerella vaginalis* were able to colonize the vagina chip, leading to alterations in epithelial cell viability, pH, lactic acid accumulation and pro-inflammatory cytokine levels [101]. A key area of research utilizing these models is dedicated to exploring the intricate interplay between invasive pathogens and vaginal microbiota. For instance, research has discovered that microbiota dominated by *Lactobacillus iners* and group B *Streptococcus* could significantly suppress the replication of human immunodeficiency virus type 1 (HIV-1) in the ALI model, while microbiota containing *Ruminococcaceae* sp., *Aerococcus* sp., *Sneathia sanguinegens*, and *Atopobium vaginae* could potentially enhance HIV-1 replication [102]. Similarly, vaginal microbiota were found to significantly alter the replication of Zika virus (ZIKV) and Herpes Simplex Virus type 2 (HSV-2), with higher levels of *S. epidermidis* associated with significantly decreased titers of both viruses [105].

Culture-based studies also provide valuable insights into pathogen–microbiota interactions. For instance, multiple strains of *L. crispatus* exhibited antibacterial activity against 11 human vaginal pathogens through the production of bacteriocins and other antimicrobial agents like lactic acid [106]. *Lactobacillus* strains isolated from vaginal swabs hindered the growth of *Escherichia coli*, *S. aureus*, *Enterococcus* species, and *Candida* species [107]. Additionally, in one study, the protozoan parasite *Trichomonas vaginalis* (TV) inhibited the growth of *L. iners* and promoted the growth of *Streptococcus agalactiae* upon initial exposure [108]. The same study also suggested that *L. iners* was capable of surviving from TV after prolonged exposure. In addition to *Lactobacillus* species, the probiotic *Lacticaseibacillus rhamnosus*, isolated from vaginal fluid, suppressed the growth of *Mycobacterium tuberculosis* in co-culture experiments, suggesting its anti-tuberculosis effect [109]. Beyond single-species isolates, cultured cervicovaginal secretions from healthy donors have shown the ability to inhibit the growth of dysbiosis-associated *Gardnerella*, making them a promising source for vaginal microbiota transplantation [110].

Other exposures studied in vitro include antibiotics, tea tree oil, and vaginal hygiene products, all of which possess antimicrobial properties. For instance, the antibiotic metronidazole effectively targeted bacterial vaginosis-associated pathogens like *G. vaginalis* and *Prevotella bivia*, but had limited impact on *A. vaginae* [103]. Moreover, the efficacy of metronidazole against *G. vaginalis* could be compromised by *L. iners*, which potentially sequestered this antibiotic [99]. Antibiotic susceptibility testing of vaginal microbiota isolates, including multiple *Lactobacillus* and *Bifidobacterium* species and *G. vaginalis*, revealed species-specific resistance traits [104,107,111]. In addition to antibiotics, tea tree oil, a potent antimicrobial compound, demonstrated fungicidal activity against multiple *Candida* strains at low concentrations (1% *v*/*v*), while minimally affecting beneficial vaginal species like *Lactobacillus* [112]. This offers a potential strategy to combat chronic vaginal *Candida* infections. However, vaginal douche products, which can suppress the growth of *Lactobacillus*, should be used with caution [113].

**Table 2 microorganisms-13-00247-t002:** Interactions between extraintestinal microbiota and environmental exposures using in vitro methods.

Exposure	In Vitro Model	Key Findings ^1^	Methodology ^2^	Reference
Respiratory tract microbiota				
Fluoroquinolone, meticillin, penicillin, oxacillin, kanamycin, tobramycin, gentamicin, erythromycin, lincomycin, tetracycline, fusidic acid, fosfomycin, rifampicin, trimethoprim/sulfamethoxazole	Nose-derived *Staphylococcus* isolates on agar plates	87 out of 88 fluoroquinolone- resistant staphylococci carried co-resistance, and 75 carried co-resistance specifically to meticillin	Disc diffusion assay	[78]
Penicillin, cefoxitin	Nose-derived *Staphylococcus* isolates on agar plates	24 out of 27 *Staphylococcus* carried resistance to penicillin and/or cefoxitin	Viable cell counting	[83]
Ampicillin, amoxicillin/clavulanate, ampicillin/sulbactam, cefuroxime, cefotaxime, imipenem, meropenem, azithromycin, tetracycline, chloramphenicol, thrimetoprim/sulfametoxazole	Throat- and nose-derived *Haemophilus parainfluenzae* isolates on agar plates	Isolates showed different resistance patterns based on two different guidelines	Disc diffusion assay	[84]
Ceftazidime, amoxicillin, cefotaxime, ceftazidime	Respiratory tract-derived *Prevotella* isolates on agar plates	38 out of 50 *Prevotella* isolates produced extended-spectrum β-lactamases and had higher resistance to amoxicillin and ceftazidime	Disc diffusion assay, Etest	[79]
Amoxicillin, amoxicillin/clavulanate, azithromycin, trimethoprim/sulfamethoxazole (TMP/SMX), ceftriaxone, levofloxacin	Nose-derived single-species isolates on agar plates	6 out of 8 *Moraxella catarrhalis* isolates carried resistance to amoxicillin and TMP/SMX, 2 of these 6 exhibited ceftriaxone resistance, and 1 exhibited azithromycin resistance 12 our of 45 *Streptococcus pneumoniae* isolates demonstrated azithromycin resistance, and 14 showed resistance to TMP/SMX	Etest	[85]
Supplemental oxygen	Sputum-derived mixed-species culture	*Candida albicans* (↓) *Aspergillus fumigatus* (↓) *Actinomyces oris* (↓) *Schaalia odontolytica* (↓) *Rothia mucilaginosa (↓)* Multiple *Streptococcus* species (↓) *Pseudomonas aeruginosa* (-) *Staphylococcus aureus* (-)	Metagenomic shotgun sequencing	[86]
Human rhinovirus (HRV)	*Corynebacterium*, *Haemophilus influenzae*, Calu-3 cell co-culture in the air-liquid interface (ALI) model	HRV copy number (↓) by *Corynebacterium pseudodiphtheriticum* + *Haemophilus influenzae*	qRT-PCR	[77]
Skin microbiota				
Nine cosmetics	*Staphylococcus epidermidis* single-species culture	Yields of short-chain fatty acids depended on different cosmetics	HPLC	[89]
Ultraviolet (UV) filters in sunscreens	*Lactobacillus crispatus*, *Staphylococcus epidermidis*, and *Cutibacterium acnes* single-species culture in a culture plate exposed to UV light	*Lactobacillus crispatus* (↑) *Cutibacterium acnes* (↓)	Viable cell counting	[90]
Octocrylene	Skin-derived single-species culture	*Deinococcus grandis* and *Stenotrophomonas* grew by metabolizing octocrylene	Optical density measurement	[91]
Ultraviolet radiation (UVR) at 254 nm or 312 nm	*Sphingomonas mucosissima* single-species culture on agar plates	*Sphingomonas mucosissima* was resistant to UVR at both wavelengths	Visual observation	[92]
Ultraviolet A (UVA), ultraviolet B (UVB), or combination	Skin-derived *Staphylococcus aureus*, *Staphylococcus epidermidis*, *Staphylococcus hominis*, *Micrococcus luteus*, *Corynebacterium stearicum*, and *Moraxella osloensis* single-species culture	UVA: complete inhibition of all microorganisms UVB: strain-dependent inhibition Combination: similar to UVA	Optical density measurement	[93]
Ultraviolet B (UVB)	*Cutibacterium acnes* and *Lactobacillus fermentum* single-species culture	Both microorganisms (↓)	Viable cell counting	[94]
Mycolactones	Skin-derived single-species fungal spores on agar plates	*Aspergillus flavus* (↑) *Aspergillus niger* (↑) *Penicillium rubens* (↓)	Germinated spore quantification	[95]
Benzo[a]pyrene	Skin-derived *Micrococcus luteus* and *Pseudomonas oleovorans* co-culture in a microbially competent 3D skin model	Benzo[a]pyrene degradation to various metabolites	GC-MS	[87]
Methyl Red, Orange II	Single-species culture	*Staphylococcus*, *Corynebacterium*, *Micrococcus*, *Dermacoccus*, *and Kocuria* species metabolized Methyl Red with various rates, and all but *Corynebacterium xerosis* metabolized Orange II	Spectrophotometry	[96]
Doxycycline, ciprofloxacin, erythromycin, cefalexin, amoxicillin, trimethoprim, clarithromycin, linezolid, metronidazole, azithromycin, co-amoxiclav	*Staphylococcus epidermidis* single-species culture on agar plates	*Staphylococcus epidermidis* exhibited resistance to various antibiotics, and antibiotic-adapted strains showed cross-resistance	Disc diffusion assay	[97]
Green tea extracts	Single-species culture on agar plates	*Micrococcus luteus* (↓) *Staphylococcus epidermidis* (↓) *Clostridium xerosis* (↓) *Bacillus subtilis* (↓)	Optical density measurement	[98]
Vaginal microbiota				
Human immunodeficiency virus type 1 (HIV-1)	Vagina-derived single species or mixed species co-cultured with vaginal epithelial cells and HIV-1-susceptible cells in the air-liquid interface (ALI) model	HIV-1 replication (↓) by *Lactobacillus iners* and group B *streptococcus*-dominated culture	qRT-PCR	[102]
Zika virus (ZIKV), Herpes Simplex Virus type 2 (HSV-2)	Vagina-derived single species or mixed species co-cultured with vaginal epithelial cells in the air-liquid interface (ALI) model	ZIKV titers (↓) by *Staphylococcus epidermidis*-dominated culture ZIKV titers (↑) by *Lactobacillus crispatus*-dominated culture HSV- HSV-2 (↑) by *Lactobacillus jensenii*-dominated, *Mobiluncus mulieris*-containing culture	qPCR	[105]
Human vaginal pathogens including *Enterococcus faecalis*, *Staphylococcus aureus*, *Staphylococcus epidermidis*, *Streptococcus agalactiae*, *Escherichia coli*, *Klebsiella pneumoniae*, *Pseudomonas aeruginosa*, *Gardnerella vaginalis*, and *Mobiluncus curtisii*	*Lactobacillus* single-species culture on agar plates	Pathogens (↓) by *Lactobacillus* species except for *L. iners*, with strain-specific differences	Zone of inhibition surrounding *Lactobacillus*	[106]
Human vaginal pathogens including *Escherichia coli*, *Staphylococcus aureus*, *Enterococcus*, and *Candida albicans*	Vagina-derived *Lactobacillus* single-species culture on agar plates	Pathogens (↓), with strain-specific differences	Spots-on-lawn test	[107]
*Trichomonas vaginalis* (TV)	*Streptococcus agalactiae* and *Lactobacillus iners* single-species culture	*Lactobacillus iners* (↓) upon TV exposure, and (-) six hours later *Streptococcus agalactiae* (↑)	Viable cell counting	[108]
*Mycobacterium tuberculosis*	Vagina-derived *Lacticaseibacillus rhamnosus* single-species culture	*Mycobacterium tuberculosis* (↓)	Viable cell counting	[109]
Gardnerella	Vagina-derived mixed-species culture on agar plates	*Gardnerella* (↓)	Zone of inhibition surrounding mixed-species culture	[110]
Metronidazole	*Lactobacillus crispatus*, *Lactobacillus iners*, *Gardnerella vaginalis*, *Prevotella bivia*, and *Atopobium vaginae* co-culture	*Gardnerella vaginalis* (↓) *Prevotella bivia* (↓) *Atopobium vaginae* (↓) *Lactobacillus crispatus* (-) *Lactobacillus iners* (-)	Optical density measurement	[103]
Metronidazole	*Gardnerella* vaginalis and *Lactobacillus iners* co-culture	*Gardnerella vaginalis* (-) due to metronidazole sequestration by *Lactobacillus iners*	Viable cell counting	[99]
Metronidazole, clindamycin	Vagina-derived *Bifidobacterium* single-species culture on agar plates	*Bifidobacterium* exhibited different susceptibility to metronidazole and clindamycin, with species-specific patterns	Etest	[104]
β-lactamines, aminoglycosides, tetracyclines, macrolides, glycopeptides, sulfamides, diaminopyrimidine, rifamycines, aminosides	Vagina-derived *Lactobacillus* single-species culture on agar plates	*Lactobacillus* showed species- and strain-dependent antibiotic resistance patterns	Disc diffusion assay	[107]
Clindamycin, erythromycin, metronidazole, tinidazole, dequalinium	*Gardnerella vaginalis* single-species culture	*Gardnerella vaginalis* showed strain-dependent antibiotic resistance patterns	Optical density measurement	[111]
Tea tree oil	Vagina-derived single-species culture	*Candida* (↓) at low oil concentration *Bifidobacterium* (↓) at intermediate concentration *Lactobacillus* (↓) at high concentration	Agar well diffusion assay	[112]
Three vaginal douche products	*Lactobacillus* single-species culture	*Lactobacillus* (↓)	Minimal inhibiting concentration measurement	[113]

^1^ The symbols (↑), (↓) and (-) represent significant increases, decreases, and no significant changes in microbial growth, abundance, or activity, respectively, as observed after exposure. ^2^ qRT-PCR: quantitative reverse transcription polymerase chain reaction; HPLC: high-performance liquid chromatography; GC-MS: gas chromatography coupled to mass spectrometry; qPCR: quantitative polymerase chain reaction.

## 4. Discussion

One benefit of current in vitro models lies in their ability to support the establishment of accurate dose–response relationships between various exposures and microbiota. This functionality can aid in clarifying the molecular mechanisms through which microorganisms influence exposures and vice versa. Such insights are particularly valuable for investigating microbial susceptibility and resistance to antimicrobial agents [43,78,79,83,84,97,98,104,107,111]. In contrast, in animal models or human subjects, exposures are frequently altered by the host prior to or during interaction with the microbiota, complicating the control of this process due to the intricate nature of host interactions. By gaining a deeper understanding of the molecular mechanisms, researchers can enhance their ability to forecast health outcomes linked to environmental exposures and devise targeted strategies to preserve healthy microbiota.

Moreover, in vitro models serve as valuable tools for screening and expanding potential probiotics aimed at reducing risks associated with various exposures. For example, *Bifidobacterium* strains isolated from healthy vaginal microbiota, traditionally thought to be predominantly composed of *Lactobacillus*, possessed the ability to produce lactic acid and withstand low pH levels, thus providing a protective role similar to that of *Lactobacillus* [104]. *L. rhamnosus*, also obtained from vaginal microbiota, could inhibit the growth of *M. tuberculosis*, thereby presenting itself as a potential candidate for anti-tuberculosis drug development [109].

Additionally, in vitro models can be personalized to rapidly investigate the unique microbiota of each individual and their responses to environmental exposures, as shown in previous studies [42,45,49]. This capability is particularly useful for precision medicine, as it enables predictions of individual reactions before clinical interventions or treatments, and eliminates concerns regarding interpersonal variability. A possible application lies within the domain of microbiota transplantation, wherein microbiota from prospective donors may be tested against pathogens obtained from particular patients to identify the most appropriate donor [110].

Finally, in vitro, lab-based models can yield data that either support or validate computational models simulating the interactions between microbiota and various exposures, as well as interactions between microorganisms. For instance, one multispecies computational model was created to simulate the transitions between BV-associated bacteria and *Lactobacillus* species following exposure to metronidazole, and the association was later validated in an in vitro co-culture setting [99].

In vitro studies have provided valuable insights into the interactions between environmental exposures and the human microbiota. However, these models often overlook the host’s role in modifying exposures before they reach the microbiota, especially those in the lower GI tract. While models like SHIME simulate digestion and enzymatic reactions, they cannot fully replicate host-mediated processes like absorption in the small intestine, limiting their ability to accurately mimic real-world exposure scenarios. Furthermore, these models do not adequately consider the metabolic interactions between the host and microbiota, nor do they incorporate other host-related elements such as immune responses and systemic feedback mechanisms. Additionally, many in vitro systems tend to oversimply conditions for microbial growth, disregarding factors such as nutrient flow, mechanical forces, and oxygen gradients that influence microbial behavior in vivo. Organ-on-a-chip technology, which integrates various cell types and microenvironments, shows promise for addressing these limitations [114]. By replicating the complex physiology of target organs, including the epithelium, immune cells, and microbiota, this technology can enable more comprehensive investigations into the interplay between host cells, microorganisms, and environmental exposures, ultimately leading to a more comprehensive understanding of exposure outcomes and effects. Nevertheless, it can be challenging to simulate long-term microbial adaptation patterns and host systemic responses on chips, which is crucial for comprehending the effects of chronic exposures.

Another limitation of current studies is the lack of research on how multiple exposures combine to affect the microbiota. In real-world scenarios, individuals encounter complex mixtures of chemicals and conditions. In vitro models are ideal for studying these combined effects, including potential interactions between exposures and their overall impact on the microbiota. Additionally, such models can generate data for developing computational tools that predict how the microbiota respond to new exposure combinations. Nevertheless, this review found very limited studies of this nature. This type of research is crucial for informing regulations and risk assessments, allowing the prioritization of the most concerning mixtures for further investigation in vivo and in clinical trials.

Furthermore, a significant number of existing studies are primarily observational or merely indicate associations, without thoroughly elucidating the mechanisms underlying microbial alterations induced by exposure. To address this gap, omics methodologies offer a comprehensive approach to investigating the intricate microbiota. Whole-genome sequencing can facilitate the identification of genetic factors influencing exposure metabolism, while RNA sequencing can be employed to examine gene expression patterns in relation to varying exposure levels. Although these techniques have been effectively applied in certain studies [45,46,53], there is a necessity for broader implementation for a deeper understanding of the complex interplay between exposure and microbiota.

Finally, in-depth in vitro studies of the stomach and small intestinal microbiota are lacking. A significant challenge is their relative inaccessibility. Unlike the colon, oral cavity, skin and vagina, which can be sampled non-invasively through swabs, the stomach and small intestine typically require invasive procedures like endoscopy or ileostomy [73,74]. These methods are costly, uncomfortable for patients, and carry a risk of contamination. A potentially less invasive approach is to adapt capsule endoscopy for collecting fluid or tissue samples for more comprehensive analysis [115,116].

## 5. Conclusions

In vitro models have emerged as valuable tools for studying the intricate interactions between environmental exposures and human microbiota. These models offer advantages over conventional animal and epidemiological studies by providing a controlled, reproducible, and ethically sound environment to investigate exposure–microbiota dynamics. While research has predominantly focused on large intestinal microbiota, growing attention to microbiota in other body regions, such as oral, gastric, small intestinal, skin, respiratory, and vaginal microbiota, has revealed promising findings. However, current limitations, including a lack of clarity surrounding host–microbiota interactions, insufficient exploration of multi-exposure effects, and a deficiency in mechanistic studies, highlight opportunities for improvement. Future advancements in organ-on-a-chip technologies, multi-exposure studies, and omics approaches will be crucial in bridging existing knowledge gaps. Ultimately, the continued development and application of in vitro models will enhance the understanding of microbiota-mediated health outcomes, paving the way for precision medicine strategies and improved risk assessment frameworks.

## Data Availability

The original contributions are presented in the article; further inquiries can be sent to the corresponding author.

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
