# Peer review of "Using In Vitro Models to Study the Interactions Between Environmental Exposures and Human Microbiota"

_microorganisms, 2025, doi:10.3390/microorganisms13020247_

Round 1

Reviewer 1 Report

Comments and Suggestions for Authors

Dear Authors,  

My main concern is that similar review papers have been performed, which analyzed the interactions between environmental exposures and human microbiota in vitro.  

Specific comments:  

In abstract, clearly define the process to literature searching (e.g., PubMed, Scopus...etc.), summarize the key findings, and connect the findings to practical implications and the potential impact on gut health.

 The most common weaknesses with this review are lack of novelty/originality of the topic, and weak description of the methods informing data. First, it is unclear what this review adds in light of several recent reviews on this topic (Microbiol Res. 2023 May:270:127336; Front Microbiol. 2022 Apr 12;13:828359; Microorganisms. 2024; 12(9):1828; Microorganisms. 2024; 12(1):67; Toxicol Sci. 2020 May 11;176(2):253–284; Curr Opin Endocr Metab Res. 2020 Oct 3;13:39–48). The introduction should more directly articulate the specific research gap this review addresses and the unique insights it offers. Why the interactions between environmental exposures and human microbiota in vivo were not explored? What were the convergence points regarding environmental exposures and human microbiota between animal and human studies?

Second, clearly organize the method steps and criteria for easy understanding. Be more explicit about how each criterion was applied, especially regarding the selection of articles. What were the inclusion and exclusion criteria? Clarify the process of data extraction and how discrepancies were resolved.

All tables are weakly presented. Need to add the methodology that was used in the individual cited publications. I would like to point out that the microbiome assessment is largely dependent on the choice of testing method, e.g. 16 SRNA, metagenomics sequencing, and others. Also, the environmental exposures should be clearly stated in all tables.

The mechanisms by which environmental exposures influence intestinal microbiota should be clearly stated.

There is need to consider incorporating a figure to summarize the key findings, which can enhance the clarity and impact of the manuscript.

There is also need to add a conclusion section with the consequences of the results obtained and recommendations for the future studies.

Avoid using the terms "we" and "our" throughout the manuscript (e.g., we will, our review, our work, our search).

Reviewer 2 Report

Comments and Suggestions for Authors

It is an interesting review showed that the  the interactions between environmental exposures and human microbiota using in vitro cell culture method. The authors discussed the interaction netween intestinal microbiota and enviornments using in vitro cell culture, the interaction netween extraintestinal microbiota and enviornments using in vitro cell culture.

Major points:

1- Mechanisms of interactions are missings

2- Effect of this interaction on disease progression and clinical settings is not clear.

3- How the interaction affect the immune system is not clear

4- What conditions favor this interaction and condition inhibit it

iota

Reviewer 3 Report

Comments and Suggestions for Authors

This is an elegant, well-done review about an important issue, the use of in vitro models to study the interactions between environmental exposures and human microbiota.

This review highlights the strong link between human microbiota and health, emphasising how environmental factors like chemical contaminants and biological agents can alter microbiota composition and function, impacting health outcomes. Traditional research methods, such as animal studies and epidemiological approaches, are often expensive, time-consuming, and ethically challenging. The review advocates for the use of in vitro models as a faster, cost-effective, and reliable alternative to study these interactions. It discusses the benefits of in vitro models, identifies knowledge gaps, and suggests directions for future research in understanding microbiota-environment interactions.

Overall, this review is a valuable contribution to the growing field of microbiome-centered research. However, to be published, in my opinion, there are some changes to be made:

1)                       Page 2: The sentence “have been instrumental in elucidating these interactions.” should be clearer. Do the authors mean “have been instruments in elucidating these interactions”? or “have been used in elucidating these interactions”?

2)                       Page 4-12: In my opinion, Table 1 should be placed at the end of Section 3, and Table 2 at the end of Section 4 (Extraintestinal Microbiota). If the authors have a rationale for the placement of Tables 1 and 2 in their current locations, it should be clearly stated. For me, it is not clear why these tables are positioned where they are and would be better to be other way.

3)                       In my opinion the main disadvantages of using in vitro models should also be discussed in more detail. Namely the lack the full biological complexity of a living organism, such as immune responses, metabolic interactions, and systemic feedback mechanisms, the controlled conditions of in vitro models may not accurately replicate the natural microenvironment, including pH, oxygen levels, and interactions with other tissues, the inability to simulate long-term effects and the limited scalability for whole-system responses

Apart from these 3 points that should be taken in consideration, this is a valuable review that has the quality to be published. 

Reviewer 4 Report

Comments and Suggestions for Authors

I am sincerely grateful for the opportunity to review this interesting and innovative text. This work significantly contributes to the fascinating field of research on the interaction between the human microbiome and environmental factors. The clarity and depth with which you address the topic are truly commendable. In particular, the discussion on the current limitations of in vitro models and the future perspectives for overcoming them is insightful.  

This research paper delves into the effectiveness of in vitro models in studying how environmental factors influence gut bacteria. The authors highlight the advantages of these models, such as speed and low cost, compared to traditional methods like animal or human studies. They emphasize the significance of understanding these interactions for human health, a crucial area that has the potential to impact our understanding of health and disease significantly. In simpler terms, scientists are using laboratory models to comprehend better how our surroundings affect our gut bacteria and, consequently, our overall health. The article presents a comprehensive overview of the topic. While the overall structure and presentation are clear, the manuscript could be improved.

In the second paragraph of the introduction, the authors use the term consumption. This term implies a voluntary act, which might not accurately reflect the involuntary nature of the interaction. It might be more appropriate to use the term 'exposure,' which better describes the involuntary nature of the interaction.

Secondly, in Figure 1, there is a discrepancy in the final number of included articles. If 753 articles were initially eligible and 191 were excluded, the final number of included articles should be 56, not 52. This discrepancy might be due to a calculation error. This issue needs to be addressed.

Table 2 is misplaced and not referenced in the text. It should ideally be positioned in section 4, closer to where it is first mentioned. Additionally, it must be clarified if Table 2 is the same as Table S1, referenced in section 4.

The statement in section 3.1 suggests that pregnancy or cardiovascular disease is a systemic health condition that requires revision. While pregnancy involves systemic changes, it is not typically classified as a disease. The sentence could be rephrased to avoid these implications.

The paragraph in section 3.2 discussing the role of gut microbiota could benefit from additional citations. The statement regarding the gut-brain axis and the production of short-chain fatty acids (SCFAs) would be strengthened with references to supporting literature.

Finally, the methodology section requires more detail. The authors should provide a more in-depth description of the article selection process, including the databases searched, search terms used, and inclusion/exclusion criteria. A summary of the key findings from the included studies would also be beneficial.

Overall, the article provides a valuable contribution to the field. By addressing the points mentioned earlier, the authors can further enhance their work's clarity, accuracy, and impact.

Round 2

Reviewer 1 Report

Comments and Suggestions for Authors

I understand this review focused on in vitro models to examine exposure-microbiota interactions, but the reader needs to get a better sense what the convergence points regarding environmental exposures and human microbiota between animal and human studies. By focusing on evidence only from in vitro it appears that the authors implicitly believe that there are unique aspects for these interactions. In fact, there are very strong evidence exist on these interactions in vivo. The authors should provide more justification for how their review will address this point. Otherwise, it does not make no sense to the reader.

It would be benefit to choose a review type suitable for the review purpose. It is vague that the authors selected 52 studies in this review. This should be changed to narrative review/traditional review not a systematic review. A systematic review aimed to provide more details on a statistical analysis and methods for evaluating heterogeneity, sensitivity analysis, publication bias. Unfortunately, these criteria have not been addressed. Also the authors should clearly organize the method steps and criteria in separate sub-section for easy understanding, and provide more details on the search strategies, specifying the Boolean operators and any additional filters used. In my opinion, search terms should be clearly defined.

It would be beneficial for the search to be updated with August to December 2024 data.

I do not believe that the mechanisms by which environmental exposures influence human microbiota were not explored, given this review focused on in vitro studies. The mechanisms underlying microbial alterations induced by exposure in some or all studies should be summarized in a separate table.

Author Response

I understand this review focused on in vitro models to examine exposure-microbiota interactions, but the reader needs to get a better sense what the convergence points regarding environmental exposures and human microbiota between animal and human studies. By focusing on evidence only from in vitro it appears that the authors implicitly believe that there are unique aspects for these interactions. In fact, there are very strong evidence exist on these interactions in vivo. The authors should provide more justification for how their review will address this point. Otherwise, it does not make no sense to the reader.

Thanks for the additional comments! We concur with the reviewer regarding the convergence observed between animal and human studies, as well as the potential for in vitro interactions to be reflected in vivo. Nevertheless, we wish to underscore that while the comparison of animal and human studies is a compelling subject, it falls outside the scope of our current work and constitutes a substantial topic that warrants a dedicated review article. In this review, our focus is solely on the insights derived from in vitro studies. Furthermore, we believe that in vitro research offers unique advantages, as it can be performed without the confounding variables associated with the host, thereby enabling a concentrated examination of exposure-microbiota interactions. Although results from in vitro studies may occasionally align with those from in vivo studies, this does not imply that in vitro findings lack value.

It would be benefit to choose a review type suitable for the review purpose. It is vague that the authors selected 52 studies in this review. This should be changed to narrative review/traditional review not a systematic review. A systematic review aimed to provide more details on a statistical analysis and methods for evaluating heterogeneity, sensitivity analysis, publication bias. Unfortunately, these criteria have not been addressed. Also the authors should clearly organize the method steps and criteria in separate sub-section for easy understanding, and provide more details on the search strategies, specifying the Boolean operators and any additional filters used. In my opinion, search terms should be clearly defined.

We have changed this review to a traditional review. We removed “a systematic review” from the title, the statement from the Methods section that this review was prepared based on the PRISMA guideline, as well as Figure 1, the flow diagram required by PRISMA. We re-organized the Methods section, and added “Search strategy”, “Eligibility criteria”, and “Data extraction and synthesis” subsections. We specified the use of Boolean operators and additional filters in Table S1. To further clarify the search process, we also added a “Keyword” column in this table to illustrate keywords.

It would be beneficial for the search to be updated with August to December 2024 data.

We updated the search and identified another six studies, three on oral microbiota [60-62], one on respiratory microbiota [85], and two on skin microbiota [93-94]. We updated the main text and Tables 1 and 2 to incorporate these new studies.

I do not believe that the mechanisms by which environmental exposures influence human microbiota were not explored, given this review focused on in vitro studies. The mechanisms underlying microbial alterations induced by exposure in some or all studies should be summarized in a separate table.

In our earlier responses, we noted that most of the current research is observational in nature and does not adequately clarify the mechanisms that drive microbial changes resulting from exposure. This limitation arises from the reliance on basic techniques, such as cell counting and optical density (refer to Tables 1 and 2), which are only capable of demonstrating correlations between microorganisms and exposures. Consequently, we believe that a table summarizing these mechanisms would not be warranted.

Reviewer 2 Report

Comments and Suggestions for Authors

The authors did not provide any answers to any of my questions.

Author Response

No updated comments are available.